# On One Solution of the Problem of Stochastic Longitudinal Oscillations of a Viscoelastic Rope With Moving Boundaries Using Artificial Intelligence and Machine Learning Methods

© Vladislav L. Litvinov

Samara State Technical University, Samara, 446001, Russia

*vladlitvinov@rambler.ru*

## ABSTRACT

Addresses the problem of stochastic longitudinal oscillations of a viscoelastic rope with moving boundaries. The main focus is on the application of artificial intelligence (AI), neural networks, and machine learning (ML) methods for analyzing resonance phenomena, predicting optimal system parameters, and preventing resonance. A method for constructing solutions to integro-differential equations is proposed, which is extended to a broader class of problems with moving boundaries. The problem of stochastic longitudinal vibrations of a viscoelastic rope with moving boundaries is formulated taking into account the influence of damping forces in the form of a system of stochastic integro-differential equations, which is reduced to the study of a system of stochastic differential equations with random initial conditions. The use of deep neural networks (DNNs), Monte Carlo methods, and adaptive control significantly improves the accuracy of predictions and the efficiency of system control. The neural network is trained on data on the behavior of the system at different frequencies and parameters. In this case, the network predicts resonant frequencies and suggests optimal parameters. The results of the AI are tested on a mathematical model. Calculations confirmed that the parameters proposed by artificial intelligence do prevent resonance.

## 1  INTRODUCTION

Systems, the boundaries of which are moving, are widespread in technology (ropes of hoisting installations [1–5, 7, 8], flexible transmission links [6], etc.). The presence of moving boundaries causes significant difficulties in describing such systems. Exact solution methods are limited by the wave equation and relatively simple boundary conditions [9]. Of the approximate methods, the most effective is the method for constructing solutions of integro–differential equations described in [7, 11, 21, 22], as well as the Kantorovich – Galerkin method [8–10, 14]. The method is extended to a wider class of model boundary value problems that take into account bending rigidity, resistance of the external environment and the rigidity of the base of the oscillating object.

The solution is implemented in dimensionless variables using the TB–Analysis software package developed in the Matlab, Python environment (libraries: TensorFlow, PyTorch, Scikit–learn), which allows using the obtained results to calculate a wide range of technical objects. The TB–Analysis software package contains a module – resonance phenomena management. The "Resonance phenomena management" command is designed to determine the resonance region and identify resonance prevention conditions imposed on the model parameters using AI.

In addition, it is recognized that deterministic modeling of systems cannot be adequate for some types of problems, so it is necessary to move on to probabilistic–statistical, where random variables and stochastic oscillations are present.

The study of viscoelasticity includes the analysis of the stochastic stability of stochastic viscoelastic systems, their reliability. Using artificial intelligence, the resonant frequencies of the system, as well as the conditions under which resonance can be prevented, were determined. The optimization of the system parameters under which the probability of resonance is minimal was carried out. Data was collected – amplitude–frequency characteristics of the system. Key parameters that affect resonance were determined. A neural network was trained on the data. In this case, the network predicts resonant frequencies and suggests optimal parameters. Nonlinear regression methods were used to determine the resonance conditions. In case of insufficient data, the data synthesis method – the Monte Carlo method – was used. ML methods were used to analyze which system parameters have the greatest influence on resonance. To determine the resonance region, frequency analysis was carried out using the fast Fourier transform (FFT) method. Clustering algorithms (k–means) were used to group frequencies at which the system is most vulnerable. Graphs of amplitude–frequency characteristics were constructed to clearly identify resonant areas. To prevent resonance, optimization of parameters was carried out

using the gradient descent method to adjust system parameters, such as increasing damping and changing rigidity. Adaptive control methods and neural networks were used for dynamic adjustment of the system in real time.

The results of the AI work are tested on a mathematical model. Calculations confirmed that the proposed parameters really prevent resonance.

This paper proposes an approach based on the application of artificial intelligence (AI) and machine learning (ML) methods, which not only improves the accuracy of calculations but also significantly reduces the time required to find optimal system parameters.

## 2 MATHEMATICAL STATEMENT OF THE PROBLEM

The differential equation of motion of mechanical objects of variable length has the form

$$U_{\tau\tau}(\xi,\tau) + L[U(\xi,\tau)] = \varphi(\xi,\tau). \tag{1}$$

We write the boundary conditions in the following form

$$Y_{ji}\left[U\left(\ell_j(\varepsilon\tau),\tau\right)\right] = F_{ji}(\tau); \tag{2}$$

$$i = \overline{1,m}; \ j = \overline{1,2}.$$

Here $U(\xi,\tau)$ – offset function; $L$ – linear homogeneous differential operator with respect to $\xi$ order $2m$ ($m \leq 2$ – positive integer); $Y_{ji}$ – linear homogeneous differential operators with respect to $\xi$; $\varphi(\xi,\tau), F_{ji}(\tau)$ – specified class functions $C$ and $C^2$ respectively; $\ell_j(\varepsilon\tau)$ – boundary motion law; $\varepsilon$ – small parameter ($\varepsilon = V / a$, $V$ – border speed, $a$ – vibration propagation speed).

The movement of the boundaries according to the law $\ell_j(\varepsilon\tau)$ corresponds to the slow movement mode.

When analyzing the resonance properties, the initial conditions are taken in the form $U(\xi,0) = U_\tau(\xi,0) = 0$.

Omitting some mathematical calculations and applying the method described in works [21, 22], we obtain the following expression for the full amplitude of oscillations corresponding to the n-th dynamic mode:

$$A_n^2(\tau) = \frac{1}{4} A_{0n}^2(\varepsilon\tau) a_n^2(\varepsilon\tau) \left\{ \left[ \int_0^\tau F_n(\varepsilon\zeta) \cos \Phi_{n1}(\zeta) d\zeta \right]^2 + \left[ \int_0^\tau F_n(\varepsilon\zeta) \sin \Phi_{n1}(\zeta) d\zeta \right]^2 \right\}, \tag{3}$$

where

$$F_n(\varepsilon\zeta) = \frac{M_n(\varepsilon\zeta)}{a_n(\varepsilon\zeta)w_n'(\zeta)}; \; \Phi_{n1}(\zeta) = w_n(\zeta) - W_n(\zeta); \; \Phi_{n2}(\zeta) = w_n(\zeta) + W_n(\zeta).$$

As an example, we consider stochastic longitudinal oscillations of a viscoelastic rope with moving boundaries, taking into account damping forces.

## 3   STOCHASTIC OSCILLATIONS OF THE ROPE WITH MOVING BOUNDARIES

The differential equation governing the longitudinal oscillations of a rope, incorporating viscoelasticity based on the Voigt hypothesis, is expressed as [7, 10, 20]:

$$U_{tt}(x,t) + 2\alpha U_t(x,t) - a^2 \left[ U_{xx}(x,t) - \int_0^t K(t-v)U_{xx}(x,v)dv + \mu U_{xxt}(x,t) \right] = f(x,t). \quad (4)$$

Boundary conditions are given by:

$$U(v_0 t, t) = 0; \; U(v_0 t + l_0, t) = 0. \quad (5)$$

Initial conditions are specified as:

$$U(x,0) = U_1(x); U_t(x,0) = 0. \quad (6)$$

To simplify the problem, new variables are introduced to fix the boundaries:

$$\xi = (x - v_0 t)/l_0; \quad \tau = at/l_0; \quad U(x,t) = V(\xi,\tau).$$

The function $F(\xi,\tau)$ can be represented as

$$F(\xi,\tau) = \sum_{n=1}^{\infty} F_n(\tau)\sin(\omega_n\xi), \; \omega_n = \pi n. \quad (7)$$

The initial conditions and external load are treated as random, modeled as a sum of sinusoids with random amplitudes, denoted as $\tilde{V}(\xi)$ and $\tilde{F}(\xi,\tau)$, respectively. As a result, the oscillations become stochastic, and equations (4) form a system of random integro-differential equations:

$$\tilde{V}_{n_{\tau\tau}}(\tau) + \left( 2k_1 + \frac{\lambda}{v}\omega_n^2 \right)\tilde{V}_{n_\tau}(\tau) + \omega_n^2(1-v^2)\tilde{V}_n(\tau) +$$

$$+\omega_n^2 d\int_\xi^1 K(-d(\xi-\eta))\tilde{V}_n\left( \frac{1}{v}(\xi-\eta) + \tau \right)d\eta = \tilde{F}_n(\tau); \quad (8)$$

$$\tilde{V}_n(0) = 2\int_0^1 \tilde{V}_1(\xi)\sin(\omega_n\xi)d\xi; \; \tilde{V}_{n_\tau}(0) = 0. \quad (9)$$

Here $\; v = \dfrac{v_0}{a}; \; d = \dfrac{l_0}{v_0}; \; k_0 = \alpha v l_0; \; k_1 = \alpha v d; \; \lambda = \dfrac{\mu}{d}; \; F(\xi,\tau) = v^2 d^2 f(x,t).$

To determine the statistical characteristics (mathematical expectation, variance, and covariance) of the stochastic linear longitudinal oscillations of a viscoelastic rope, it is necessary to derive statistical estimates for the solution of the system of random integro-differential equations (8). For this purpose, the relaxation kernel $K(z)$ can be expressed in an exponential form with a random component:

$$K(z,\overline{\beta}) = K(z,\overline{b})\big|_{\overline{b}=\overline{\beta}} = \sum_{j=1}^{N} c_j e^{-\beta_j z}, \tag{10}$$

where $c_j \in R_+$, $\beta_j$ – is a possible value of a positive random variable $b_j$.

Denote the dependence of $\tilde{V}(\xi,\tau)$ and $\tilde{V}_n(\tau)$ on the random vector $\overline{b}$ as $\tilde{V}(\xi,\tau,\overline{b})$ and $\tilde{V}_n(\tau,\overline{b})$, respectively. By changing the variable

$$u_{nj}(\tau,\overline{b}) = \int_{\xi}^{1} e^{-\beta_j d\eta} \tilde{V}_n\left(\frac{1}{v}(\xi-\eta)+\tau\right),\overline{b})d\eta \tag{11}$$

the system of random integro-differential equations (8) is converted into a system of random differential equations of the form:

$$\tilde{V}_{n_{\tau\tau}}(\tau,\overline{b}) + \left(2k_1 + \frac{\lambda}{v}\omega_n^2\right)\tilde{V}_{n_\tau}(\tau,\overline{b}) + \\ +\omega_n^2(1-v^2)\tilde{V}_n(\tau,\overline{b}) + \omega_n^2 d\sum_{j=1}^{N} c_j e^{b_j d\tau} u_{nj}(\tau,\overline{b}) = \tilde{F}_n(\tau). \tag{12}$$

The initial conditions are then expressed as:

$$\tilde{V}_n(0,\overline{b}) = 2\int_0^1 \tilde{V}_1(\xi)\sin\left(\omega_n \xi\right)d\xi; \quad \tilde{V}_{n_\tau}(0,\overline{b}) = 0; \quad u_{nj}(0,\overline{b}) = 0. \tag{13}$$

The analysis of system (12)–(13) can be performed using the statistical numerical Monte Carlo method [17–19], implemented within the «TB–Analysis» software package [23].

# 4 APPLICATION OF ARTIFICIAL INTELLIGENCE AND MACHINE LEARNING

## 4.1 Use of Deep Neural Networks (DNNs)

Deep neural networks (DNNs) are used to predict resonant frequencies and oscillation amplitudes. The network architecture includes several hidden layers with ReLU (Rectified Linear Unit) activation functions:

$$\mathrm{ReLU}(x) = \max(0, x).$$

The input data for the network includes system parameters:

- Rope stiffness $k$,

- Damping $c$,

- Boundary velocity $v$,

- External disturbances $f(x,t)$.

The output data are the resonant frequencies $\omega_n$ and amplitudes $A_n$. The network is trained using the backpropagation method and the Adam optimizer:

$$\theta_{new} = \theta_{old} - \eta \cdot \frac{\partial L}{\partial \theta},$$

where $\theta$ are the network parameters, $\eta$ is the learning rate, and $L$ is the loss function.

## 4.2 Monte Carlo Method for Estimating Stochastic Parameters

To account for random disturbances $\xi(x,t)$, the Monte Carlo method is used. Random variables are modeled using a normal distribution:

$$\xi(x,t) \sim N(0,\sigma^2),$$

where $\sigma$ is the standard deviation. The Monte Carlo method allows estimating the expected mathematical expectation, variance and covariance of the oscillation amplitude:

$$M\left(\tilde{V}(\xi,\tau)\right) = \sum_{n=1}^{\infty} M\left(\tilde{V}_n(\tau)\right)\sin\left(\omega_n\xi\right);$$

$$D\left(\tilde{V}(\xi,\tau)\right) = \sum_{n,k=1}^{\infty} D_{n,k}(\tau)\sin\left(\omega_n\xi\right)\sin\left(\omega_k\xi\right);$$

$$C\left(\tilde{V}(\xi,\tau,\zeta,\upsilon)\right) = \sum_{n,k=1}^{\infty} C_{n,k}(\tau,\upsilon)\sin\left(\omega_n\xi\right)\sin\left(\omega_k\zeta\right).$$

## 4.3 Adaptive Control Using AI

For real-time dynamic adjustment of system parameters, adaptive control based on neural networks is used. The control signal $u(t)$ is generated as:

$$u(t) = \mathrm{NN}_u\left(\mathbf{p}(t),\mathbf{y}(t)\right),$$

where $\mathbf{p}(t)$ are the current system parameters, and $\mathbf{y}(t)$ are the measured values of amplitude and oscillation frequency. The neural network $NN_u$ is trained to minimize the loss function:

$$L_{control} = \sum_{t=1}^{T} \left( A_n(t) - A_{allow} \right)^2.$$

## 4.4 Example of Using Neural Networks for Resonance Prediction

Consider an example of using a neural network to predict resonant frequencies. Let the initial parameters of the system be given in dimensionless form:

- Rope stiffness: $k_0 = 100$,

- Damping: $c = 0.05$,

- Boundary velocity: $v_0 = 0.1$.

The neural network predicts a resonant frequency $\omega_n = 5$ and an amplitude $A_n = 0.2$. The allowable amplitude is $A_{allow} = 0.1$.

1. Calculate the loss function:

$$L_{res} = \left( 0.2 - 0.1 \right)^2 = 0.01.$$

2. Calculate the gradient:

$$\nabla_{\mathbf{p}} L_{res} = 2 \left( 0.2 - 0.1 \right) \frac{\partial A_n}{\partial \mathbf{p}}.$$

3. Update the parameters:

$$\mathbf{p}_{new} = \mathbf{p}_{old} - \eta \nabla_{\mathbf{p}} L_{res}.$$

After several iterations, the system parameters are optimized, and the oscillation amplitude is reduced to the allowable level.

## 4.5 Use of Recurrent Neural Networks (RNNs)

For analyzing time series data, such as oscillation amplitude and frequency, recurrent neural networks (RNNs) are used. The RNN architecture includes hidden layers with long short-term memory (LSTM) units, which allow accounting for temporal dependencies in the data. The loss function for RNNs is: $L_{RNN} = \sum_{t=1}^{T} \left( y_t - \hat{y}_t \right)^2$,

where $y_t$ are the true values, and $\hat{y}_t$ are the predicted values.

## 4.6 Clustering Methods for Analyzing Resonance Regions

To group frequencies at which the system is most vulnerable, clustering methods such as k-means are used. The k-means algorithm minimizes the loss function:

$$L_{cluster} = \sum_{i=1}^{k} \sum_{x \in C_i} \|x - \mu_i\|^2,$$

where $C_i$ are the clusters, and $\mu_i$ are the cluster centers.

## 4.7. Use of Reinforcement Learning Methods

For real-time dynamic adjustment of system parameters, reinforcement learning (RL) methods are used. The agent is trained to maximize the reward: $R = \sum_{t=1}^{T} r_t,$

where $r_t$ is the reward at step $t$. The reward is defined as: $r_t = -\left( A_n(t) - A_{allow} \right)^2.$

## 5   SOFTWARE PACKAGE TB–ANALYSIS

The developed software package "TB–Analysis" is designed to solve a certain class of one–dimensional boundary value problems with moving boundaries, as well as for mathematical modeling and analysis of the resonant properties of objects whose states are described by these boundary value problems. The software package also allows for the selection of model parameters to prevent resonance phenomena using artificial intelligence (AI). The software was developed in the Matlab, Python environment (libraries: TensorFlow, PyTorch, Scikit–learn) environment as a standalone application. The results of Chapter 4 have been incorporated into the software package.

The user interface of the "TB–Analysis" software package consists of four windows, one of which is the startup window. This window appears when the program is launched, and the other windows can be accessed from the startup window. Additionally, these windows can be accessed from one another via a menu. The startup window of "TB–Analysis" contains three active buttons with schematic illustrations, which launch the following main modules of the software package:

1.Investigation of solutions to model boundary value problems;

2.Analysis of resonant properties of models;

3. Management of resonance phenomena.

Next to the buttons that launch the modules, the startup window provides explanations of the content and functionality of each module.

The "Management of Resonance Phenomena" command is designed to determine the resonance region and identify conditions for preventing resonance by imposing constraints on model parameters using AI.

Computations are performed using three methods for solving boundary value problems: an analytical method of variable substitution in a system of functional–difference equations [16], an asymptotic method for constructing solutions to homogeneous integro–differential equations and systems of ordinary differential equations describing the motion of objects with variable length [21, 22] and an approximate method for constructing solutions to integro–differential equations of motion for mechanical objects with moving boundaries, as described in this work.

The intelligent selection of the method depends on the analyzed model (class of integro–differential equation, initial and boundary conditions).

The computation of solutions to boundary value problems using the analytical method is implemented in the internal function "TBNumAnal."

The computation using the asymptotic method is implemented in the internal function "TBAsym."

The computation using the approximate method for constructing solutions to integro–differential equations also utilizes the results of this work, which are implemented in the function "TBNum."

An example of the module's operation is shown in Figure 1.

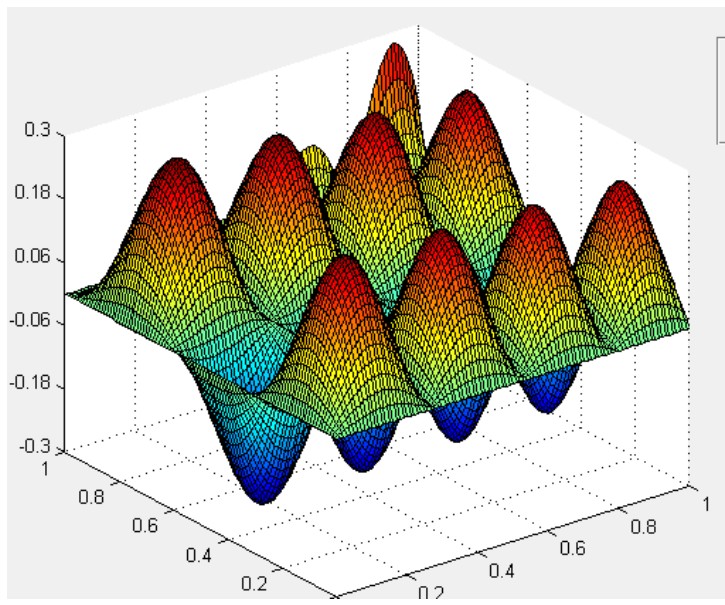

Fig. 1. Graph of the solution function for a boundary value problem
of transverse vibrations of a cable.

The expression for the amplitude of the system's oscillations in the n-th dynamic mode when passing through resonance has the following form:

$$A_n^2(\tau_1, \tau_2) = E_n^2(\tau_2) \left\{ \left[ \int_{\tau_1}^{\tau_2} F_n(\zeta) \cos \Phi_n(\zeta) d\zeta \right]^2 + \left[ \int_{\tau_1}^{\tau_2} F_n(\zeta) \sin \Phi_n(\zeta) d\zeta \right]^2 \right\}. \quad (17)$$

The described algorithm for numerical investigation of steady–state resonance and the phenomenon of passing through resonance is implemented in the function "met_ampl_max." Figure 2 shows a graph of the dependence of the maximum amplitude of cable oscillations when passing through resonance on the speed of boundary movement for various values of the environmental resistance coefficient.

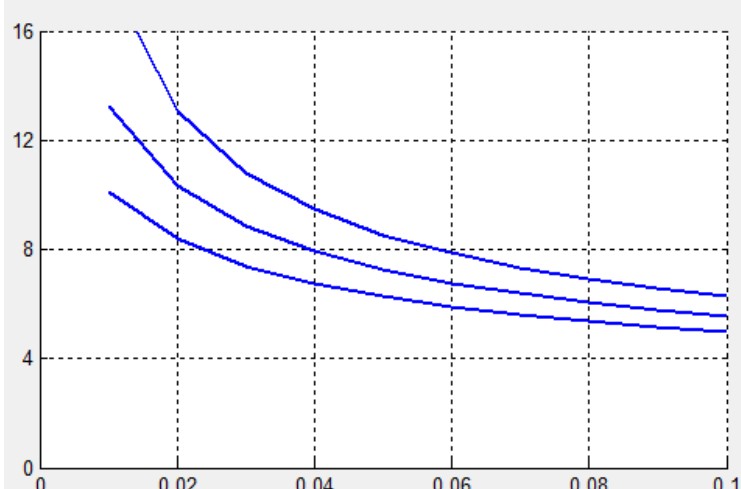

Fig. 2. Graph of the dependence of the maximum amplitude on the speed of boundary movement for various values of the environmental resistance coefficient.

In the internal function "met_ampl," which is a sub–function of the function "met_ampl_max," the computation of the oscillation amplitude expression (17) is implemented as a function of time. The results of "test" computations of the amplitude of transverse vibrations of a variable–length cable when passing through resonance on the first dynamic mode, with given initial model parameters, are illustrated in Figure 3.

In addition to their functional purpose, the graphs presented in Figure 3 also illustrate the characteristics of the oscillation amplitude behavior, which form the basis of the methodology for calculating the maximum amplitude.

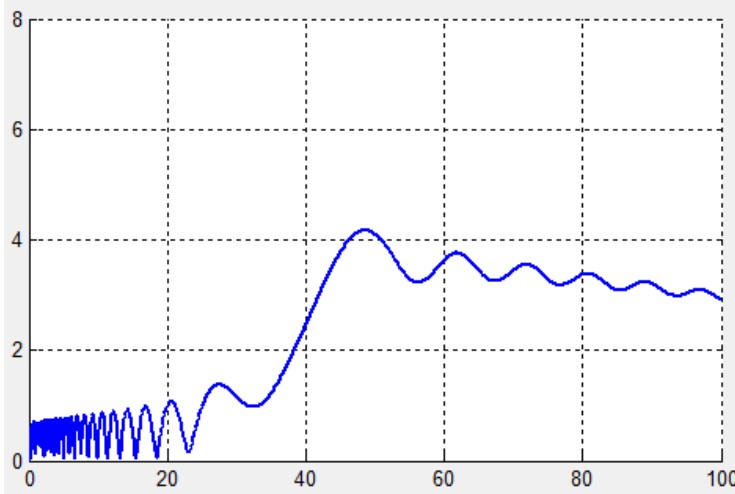
Fig. 3. Graph of the dependence of amplitude on time.

## 6   CONCLUSION

Thus, with the help of the software package "TB–Analysis" artificial intelligence was utilized to identify the resonant frequencies of the system and the conditions under which resonance could be avoided. The system's parameters were optimized to reduce the probability of resonance. Data on the system's amplitude–frequency characteristics were collected, and key parameters affecting resonance were determined. A neural network was trained on this data to predict resonant frequencies and amplitudes. Nonlinear regression methods were applied to define resonance conditions, and the Monte Carlo method was used for data synthesis when data was insufficient. Machine learning techniques were employed to analyze which system parameters most significantly influence resonance. Frequency analysis using the Fast Fourier Transform (FFT) was conducted to identify resonance regions, and clustering algorithms (k–means) were used to group frequencies where the system is most vulnerable. Amplitude–frequency characteristic graphs were plotted to clearly delineate resonance areas. To mitigate resonance, system parameters such as damping and stiffness were optimized using gradient descent. Adaptive control methods and neural networks were implemented for real–time system tuning. AI was also used to forecast conditions that could lead to resonance and to prevent it.

The problem of stochastic longitudinal vibrations in a viscoelastic cable with moving boundaries, considering damping forces, was formulated as a system of stochastic integro–differential equations. This system was simplified to a set of stochastic differential equations with random initial conditions. The Monte Carlo method was proposed for estimating expansion coefficients. The neural network was trained on data reflecting the system's behavior across various frequencies and parameters,

enabling it to predict resonant frequencies and recommend optimal parameters. The AI's predictions were validated using a mathematical model, confirming that the suggested parameters effectively prevent resonance.

The use of deep neural networks, Monte Carlo methods and adaptive control not only improves the accuracy of forecasting, but also significantly reduces the time required to find the optimal system parameters.

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
