# OpenReview forum: "ON ONE SOLUTION OF THE PROBLEM OF STOCHASTIC LONGITUDINAL OSCILLATIONS OF A VISCOELASTIC ROPE WITH  MOVING BOUNDARIES USING ARTIFICIAL INTELLIGENCE AND MACHINE LEARNING METHODS"
_mathai.club/MathAI/2025/Conference — MathAI 2025 Oral_

### Official Review · Reviewer_zkaA · 2025-02-24
**Overall assessment: Accept in its current form**

**Rating:** 7
**Confidence:** 4

**Review:**

The material of the article is original. The application of the results is possible even wider than only in oscillation problems, since similar problems should be solved in logistics, economics and dynamic modeling of network commercial structures. The structure of the article reflects the results of the work and is logically connected. The authors' arguments are substantiated and supported by a review of sources, literature and recent scientific research. It is possible to note the design of the work, good mathematics and clearly disclosed questions. Also, the results of the study of the system using the statistical numerical Monte Carlo method are presented and illustrative diagrams are presented. The number of references is sufficient to conclude that the topic has been seriously developed. The references cited in this manuscript are appropriate and relevant to this study.

---

### Official Review · Reviewer_1NjM · 2025-02-25
**Accept in its current form**

**Rating:** 7
**Confidence:** 4

**Review:**

The material of the article is original.

Remarks.
A more practical problem could be considered as an example of proposed method or comment it.
It was possible to add a few words about how proposed method would work in a multidimensional case.

The references cited in this manuscript are appropriate and relevant to this study.
The article can be accepted in present form.

---

### Official Review · Reviewer_zb55 · 2025-02-25
**There is a lot about solving the problem of oscillations of objects with moving boundaries, but the role of AI is not fully considered.**

**Rating:** 6
**Confidence:** 4

**Review:**

One of the key problems with this paper is its large size - 21 pages. This violates the requirements for the design of papers. At the same time, the use of AI is discussed only in Section 5. However, this section does not discuss mathematical methods, but the use of a certain program that somehow uses AI to estimate parameters. It remains unclear what exactly this AI is based on. If it is a neural network model, then what is its architecture? What data is used to train the AI ​​model?

Most of the article is devoted to a mechanical problem related to viscoelasticity. It considers a one-dimensional object (a rope or a rod). This does not correspond to the topic of the conference.

---

### Decision · Program_Chairs · 2025-03-08

**Decision:**

Accept (Oral)

**Comment:**

Your article has been accepted and you can make a presentation on the article. All articles will be sorted by rating and within the available conference places one author from each article will be invited. If there are not enough places, then you will either have the opportunity to present remotely or come at your own expense!